# OpenReview forum: "Mechanics of Bias and Reasoning: Interpreting the Impact of Chain-of-Thought Prompting on Gender Bias in LLMs"
_ICLR.cc/2026/Workshop/AFAA — AFAA 2026 Oral_

### Official Review · Reviewer_CjS9 · 2026-02-11
**Review: for "Mechanics of Bias and Reasoning: Interpreting the Impact of Chain-of-Thought Prompting on Gender Bias in LLMs"**

**Rating:** 4
**Confidence:** 4

**Summary:**

This paper investigates whether Chain-of-Thought (CoT) prompting meaningfully reduces gender bias in LLMs by combining benchmark evaluation across four MCQA datasets and five models, mechanistic interpretability via a Stereotype Attention Score (SAS) and hidden state probing, and qualitative analysis of reasoning chains. The authors find that CoT does not consistently mitigate gender bias at the benchmark level, that any reductions in biased attention are localized to specific head clusters while bias information persists throughout hidden representations, and that reasoning chains frequently exhibit failure modes such as dissociation, hallucination, and task hacking.

**Strengths:**

This paper appropriately positions itself within the literature, addressing important themes of bias mitigation, mechanistic interpretability, and prompt-based reasoning, which are all relevant to this workshop. The research question is whether CoT produces genuine internal change or merely superficial behavioral shifts, which is important and timely given the growing use of CoT as a low-cost intervention.

The multi-method approach that combines benchmarks, attention analysis, probing, and qualitative investigation, is the paper's strongest contribution. The experiments, including choices of datasets and metrics, are technically sound. The qualitative taxonomy of reasoning failures (Table 5) is also a valuable contribution: the six identified behaviors (dissociation, task hacking, hallucination, external sources, bias statements, reasoning correctness) are well-defined and illustrated with concrete examples. The finding that biased answers often stem from reasoning failures rather than biased reasoning per se is a nuanced insight.

The controlled comparison across the Qwen family (7B, 32B, QwQ) is a thoughtful design choice that isolates scale and training effects while holding architecture constant. The extensive appendix adds transparency and reproducibility details.

**Weaknesses:**

### Weaknesses

**Insufficient discussion of limitiations.** While the paper conducts extensive experiments, the limitations of those experiments are insufficiently addressed. The limitations section is relegated to a short paragraph in the appendix (A.6) rather than appearing in the main paper, and it does not adequately engage with several important concerns. Only four layers per model are probed (two high-SAS, one low-SAS, one random), and probes for Qwen-32B on two datasets were excluded due to class imbalance. What is lost by exploring only four layers and a subset of models? The coverage feels narrow relative to the scope of the claims being made, particularly the assertion that "gender bias information is encoded throughout a model's hidden representations." A more systematic layer-by-layer analysis, or at minimum a discussion of how the sparse sampling might affect conclusions, would help.

**Insufficient discussion of practical implications.** The paper demonstrates that bias persists in internal representations even when CoT improves output-level metrics, but does not sufficiently discuss whether this matters in practice. If the final output is bias-free (or less biased), does it matter for practitioners what happens under the hood? The paper would benefit from a more careful discussion of when internal representations matter versus when output-level evaluation suffices.

**Unexplained qualitative sampling.** How were the 257 reasoning chains selected for manual review? The paper does not describe the sampling strategy, whether random, stratified by transition category, or selected for diversity. This omission raises concerns about selection bias in the qualitative findings. Additionally, inter-annotator agreement is not reported despite four annotators participating.

### Minor Issues

**Unexplained boundary results.** The Diff-Bias score of -1 for Qwen-32B on BBQ (Table 1) in both standard and CoT conditions is an extreme result indicating the model selected the anti-stereotypical answer for every non-abstaining response. This is never discussed. It would be helpful to address this boundary result, as it could indicate overcorrection, a data artifact, or a computation issue.

**SAS heatmap readability.** The SAS heatmaps (Tables 2–4) are central to the attention analysis, but they are very difficult to read at the printed resolution and do not add as much insight as the text discussion suggests. It is difficult to connect the described patterns (specific cluster locations, sign reversals) to the visuals.

**Superficial issues.** Below is a list of simple superficial fixes I would recommend.

 - [L50] "Shaikh et al., 2022" should be \citet{} instead of \citep{}
 - [L163] Since positions are integers, the interval notation $i \in \[x_0, x_n\]$ is confusing. Suggest replacing with something closer to $x_0,..., x_n$
 - [L171] On "...subset of prompts $C$ from $M$" -> should be $D$ instead of $M$?
 - Table 1: inconsistent rounding. Recommend consistent rounding across metric results
 - [L354] "Skean et al. (2025)" should be \citep{} instead of \citet{}
 - [L374] "Kats & Belinkov (2023)" should be \citep{} instead of \citet{}

### Questions for Authors
1. What is lost by selecting only four layers in the probe analysis?
2. How were the 257 reasoning chains sampled?
3. Can you explain the Diff-Bias of -1 for Qwen-32B on BBQ?
4. If CoT successfully reduces biased outputs, what is the practical significance of bias persisting in hidden representations? Under what deployment scenarios would this internal persistence matter?

---

### Official Review · Reviewer_z28p · 2026-02-14

**Rating:** 4
**Confidence:** 4

**Summary:**

Authors explore the influence of CoT on gender bias in LLMs, leveraging "gender" subsets of well-known fairness benchmark datasets.
The paper also adopts mechanistic interpretability techniques to quantify, locate, and explain gender bias.
Authors find that CoT does not actually improve gender bias.

**Strengths:**

An interesting contribution of the paper is working at the intersection of fairness and explainability, i.e., leveraging mechanistic interpretability as a tool to discover and explain gender bias.

The focus on CoT is timely and relevant, given the latest LLM wave.

Results are clearly organized around RQs that logically follow each other and guide the reader in discovering various aspects of the relationship between gender bias and CoT.

**Weaknesses:**

An explicit definition for gender bias is missing, which would inform and better frame the contribution.
Along similar lines, at the end of page 3, a Stereotype Attention Score is introduced: stereotypes and biases are different, although related, phenomena, and they both need a clear definition. Are you measuring stereotypes or gender bias, or in the scope of the contribution they are the same thing?

Clarify better the choice of reformatting the bias benchmarks as MCQA.

Since the methodology rightfully relies on previous strategies, consider being clearer in highlighting the modifications proposed, i.e., the novel contribution that the paper brings.

The Results and Introduction section could both benefit from a paragraph stating the RQs.

Consider for each RQs bolding "the answer"/main finding.
The strength and novelty of the findings could benefit from being more clearly framed w.r.t. previous literature.

Minor aspect: introduce the MCQA acronym.

---

### Official Review · Reviewer_VK9i · 2026-02-22
**Mechanics of Bias and Reasoning: Interpret-Ing the Impact of Chain-of-thought Prompting on Gender Bias in LLMs**

**Rating:** 4
**Confidence:** 4

**Summary:**

In this paper, Chain-of-Thought (CoT) prompting is investigated to determine whether it does, in fact, mitigate (gender) biases that can be encoded in large language models (LLMs). What is found is that CoT does not consistently reduce the bias gap, as bias pervades through hidden representations.  CoT is found to fail in consistently increasing abstention rates and is contingent on model characteristics and model design. The paper does not focus solely on performance benchmarks, but rather focuses on a combination of benchmark-based evaluation with mechanistic interpretability techniques and qualitative analysis of reasoning outputs.

**Strengths:**

I found the paper to be engaging. This is not specifically my primarily area of expertise, and it was quite readable. It also is an interesting blend of methods that would be of interest for presentation discussion.
The motivation is clear from the abstract. I did find it interesting that the introduction says that this was the first such evaluation, but the abstract doesn’t mention this.
The contributions are quite clear and the paper does a great job of explicitly stating these.
I liked that the related work was split into sections that included an NLP and LLM background, with works that I recognized. The Zero-shot prompt and CoT methods drive this work as an advancement to the debate. The following section tells us what mechanistic interpretability gets us. It provides an insight into the underlying model behavior to uncover internal mechanisms in a way that is human-understandable.
This section is also the first time the paper indicates that they are extending attention-head metrics and probing classifier methods as the tool to detect the impact of CoT on how gender-bias information is represented within the model.

The figure (1) explaining methods for model extraction is readable and shows a hybrid-mechanistic and quantitative model. The question posed is also a nice example of what the model hopes to extract for further evaluation as one that can encode gender bias.
The prompting task section (3.1) does a really nice job of explaining why abstention is the best answer for not encoding gender-specific bias in the prompting tasks. The evaluation seems fair; they compare model performance in the (1) Standard and Zero-Shot CoT and. This is driven by the “step by step” approach by Kojima et al (2022).
I have to admit that the SAS insights are a bit outside of the scope of my knowledge, so I cannot evaluate this to the detail of someone with a qualitative background (e.g. who has done statistical research in psychology) that includes SAS, even though I have used Likert scale tools before.
The paper’s evaluation seems to follow that of papers such as by Zhang et al (2025) and Kojima et al. (2022).
The performance metrics seem adequate. I found that Table 1 (pg 5) was easy to interpret.

**Weaknesses:**

This is not really a weakness, but I wanted to mention that I did not have any particular issues with the paper. The only question I had was that in section 5.1, it is noted that “across models, StereoSet and CrowS-Pairs have overall lower accuracies…”. However, in section 5.2, the paper focuses on StereoSet explicitly, and I was interested as to why CrowS-Pairs was not also included in this section. I noted that the evaluation in 5 is based on StereoSet. It might have helped to also look at CrowS-Pairs as well.
Overall, the paper is an interesting read and is a good pick for discussion with the broader community and that makes it interesting.

---

### Meta-Review · Area_Chair_VYgT · 2026-02-24

**Recommendation:** Main Papers Track
**Confidence:** 4

**Metareview:**

Three reviewers have evaluated the paper with overall positive conclusions (summary assessment: 3xaccept).

The reviews particularly praise the mixed methods approach of the paper which combines benchmark evaluations and qualitative analyses. The paper is further valued for its sound experimental setup, including the choice of datasets and metrics, which eventually leads to novel and nuanced insights. The paper is well positioned in the related literature and clearly a good fit for the workshop theme.

The concerns that are raised can largely be addressed by minor revisions, which include a more elaborated discussion of limitations and practical implications in the main text, next to methodological clarifications and polishing.

Based on the reviews, I propose the paper to be accepted for the workshop.

---

### Decision · Program_Chairs · 2026-03-02

Accept (Oral)